# Understanding challenges as they impact on hospital-level care for pre-eclampsia in rural Ethiopia: a qualitative study

Tanya Robbins ,[1] Andrew Shennan,[2] Jane Sandall,[1] Tigist Eshetu Guangul,[3] Rahel Demissew,[4] Ahmed Abdella,[4] Rosie Mayston,[5] Charlotte Hanlon [3,6]

For numbered affiliations see end of article.

**Correspondence to**
Dr Tanya Robbins;
tanya.robbins@kcl.ac.uk

## ABSTRACT

**Objective** To explore hospital-level care for pre-eclampsia in Ethiopia, considering the perspectives of those affected and healthcare providers, in order to understand barriers and facilitators to early detection, care escalation and appropriate management.

**Setting** A primary and a general hospital in southern Ethiopia.

**Participants** Women with lived experience of pre-eclampsia care in the hospital, families of women deceased due to pre-eclampsia, midwives, doctors, integrated emergency surgical officers and healthcare managers.

**Results** This study identified numerous systemic barriers to provision of quality, person-centred care for pre-eclampsia in hospitals. Individual staff efforts to respond to maternal emergencies were undermined by a lack of consistency in availability of resources and support. The ways in which policies were applied exacerbated inequities in care. Staff improvised as a means of managing with limited material or human resources and knowledge. Social hierarchies and punitive cultures challenged adequacy of communication with women, documentation of care given and supportive environments for quality improvement.

**Conclusions** Quality care for pre-eclampsia requires organisational change to create a safe space for learning and improvement, alongside efforts to offer patient-centred care and ensure providers are equipped with knowledge, resources and support to adhere to evidence-based practice.

## INTRODUCTION

Poor-quality maternal care in hospitals contributes significantly to avoidable deaths. Pre-eclampsia is a leading cause of maternal and perinatal mortality and morbidity globally.[1] The greatest burden is borne by low-income and middle-income countries (LMICs).[2] There is a dearth of robust national data on incidence, particularly in LMICs. It is estimated that 2%–5% of pregnancies globally are affected by pre-eclampsia.[2 3] In Ethiopia, 19% of maternal deaths are attributed

to hypertensive disorders of pregnancy (HDPs) compared with 14% worldwide.[1 4 5] In addition to these uncertain estimates, there is an increasing burden of non-communicable diseases in LMICs.[6 7] These include chronic hypertension and diabetes, which are known to increase the risk of developing pre-eclampsia. Good quality care makes a difference to outcomes; the majority of deaths due to pre-eclampsia are preventable. Less than one in a million women delivering in the UK now dies due to pre-eclampsia, compared with an estimated 783 deaths per million in Ethiopia.[8]

Internationally, there is a lack of consensus concerning classification, diagnosis and management of pre-eclampsia.[9] Controversy remains over inpatient versus outpatient management, target blood pressure (BP) and timing of delivery for preterm pre-eclampsia.[10] However, there is evidence to support prophylaxis with low-dose aspirin and supplemental calcium if dietary intake is low, treatment of severe hypertension ($\geq$160/100 mm Hg) and use of magnesium sulfate for eclampsia or pre-eclampsia with severe features.[11–16] Prediction and prevention of pre-eclampsia is an important goal in reducing mortality and morbidity. However,

robust health systems and continuity across different levels of the care pathway are needed to support effective implementation of these interventions.

Recent proposals for rethinking models of maternity care to improve quality and equity have called for increased focus on centralised services and shifting all deliveries to or near hospital settings.[17] Justifications included the limitations of obstetric risk stratification, low-quality care in primary level settings and challenges of referral if complications do occur. However, to address equity, redesign must be country-led, context-specific and codesigned with a focus on listening to women and strengthening human resources.[18] As poor-quality care emerges as a greater barrier to reducing mortality than insufficient access to services, understanding challenges in secondary care and the experiences of women cared for in hospital is critical.[19] A large, international prospective dataset showed that 61% (28.1% within a central referral hospital) of eclampsia cases in Ethiopia occurred within facilities, highlighting the need to improve quality of care as well as surveillance, early detection and access to appropriate care.[20]

The objective of this study was to develop rich contextual understanding of the processes and experiences of hospital-level care for pre-eclampsia in rural Ethiopia to inform efforts to improve quality.

## METHODS
### Study setting
This study was conducted in a primary and general hospital in the Gurage Zone, Southern Nations, Nationalities and Peoples' Region (SNNPR). This study setting was selected to include a predominantly rural population, spread across lowland and highland areas. This is to provide generalisability of our findings. Geographical barriers in the area are significant and access to ambulances limited. Ethiopia has an ethnically diverse population of over 112 million, 79% of whom live in rural areas. Government-run health services are structured into three-tiers: primary, secondary and tertiary care (see figure 1). The general hospital at secondary level care serves as a referral centre for the primary hospital. In the study setting, maternity services at the primary hospital were provided by midwives, junior doctors referred to as general practitioners (GPs) and integrated emergency surgical officers (IESOs). IESOs are non-medical clinical staff trained to provide emergency surgical care. In addition to these cadres, the general hospital was also staffed by an obstetrician.

Ethiopia has seen significant reductions in maternal mortality ratios over recent decades from 1030 deaths per 100 000 live births in 2000 to an estimated 401 in 2017.[21] However, institutional delivery rates remain low, estimated

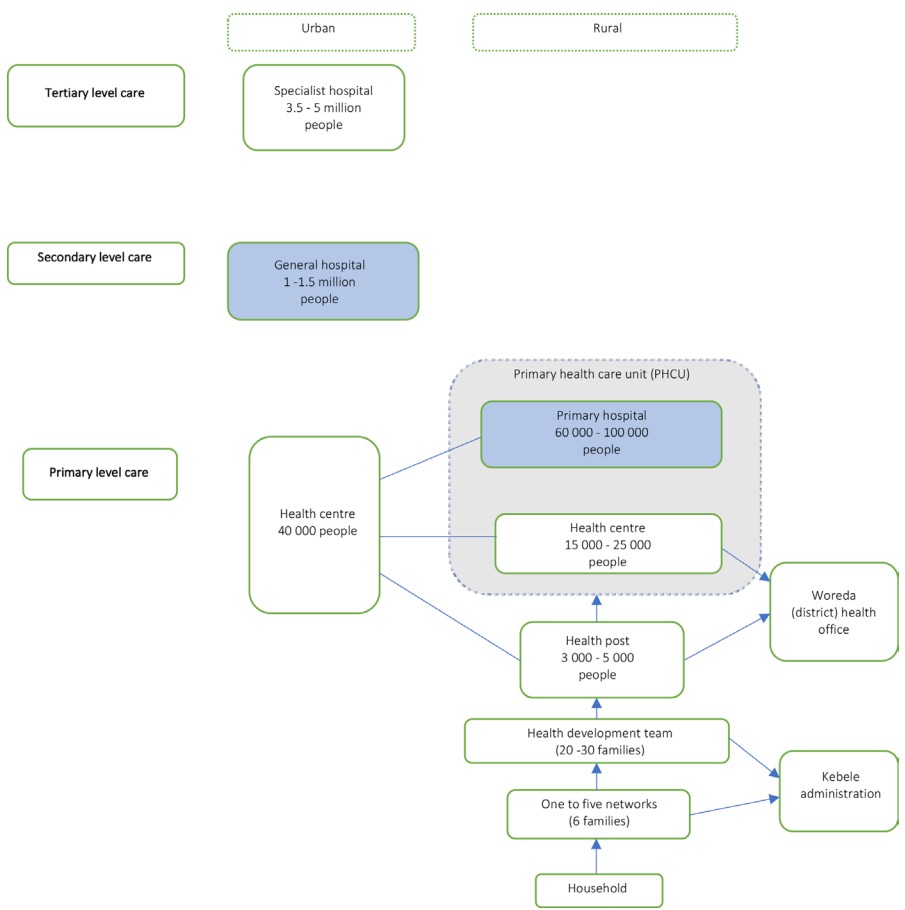

**Figure 1** Ethiopian health system structure.

at 26% in 2016, and only 43% of pregnant women had four or more antenatal care (ANC) visits in 2019. The Ethiopian Federal Ministry of Health (FMoH) produces strategies to improve all levels of the health system set out in 5 yearly Health Sector Transformation Plans to guide healthcare delivery.[22] These align with current international commitments such as achieving quality, person-centred, respectful and equitable healthcare.

## Data collection

We conducted a qualitative study between January 2019 and January 2020 using unstructured participant observation, semistructured interviews and document analysis of hospital guidelines and meeting minutes. Our research team included both an Ethiopian (RD) and a British obstetrician (TR) and an Ethiopian social epidemiology researcher (TEG). Two researchers conducted rapid ethnographic observation over 16 days between the primary and general hospitals.[23] Observers included a clinician and an Amharic speaker at all times. We observed all aspects of maternity care relevant to pre-eclampsia including spending time on the labour ward, in the emergency department, antenatal clinic, inpatient wards and in the operating theatre. Ethical challenges about when and how to intervene when questionable practices were observed were discussed between clinical members of the research team. Clinical researchers assisted in life-threatening situations. Where staff explicitly asked for assistance with non-critical cases they were encouraged to continue with routine practice. Informal interviews were conducted with staff to enquire about specific aspects of care or interactions during participant observation where further clarification or context was sought. Contemporaneous field notes were taken and discussed between researchers encouraging reflexive practice. Twenty-one formal in-depth semistructured interviews were conducted. We purposively sampled 10 staff members (2 midwives, 3 IESOs, 4 doctors including an obstetrician and a district health manager), 8 women with lived experience of pre-eclampsia and three family members of women who had died due to pre-eclampsia. Women affected by pre-eclampsia were identified through routinely collected data. The research team conducting interviews were not known to interviewees. Interviews were conducted in Amharic, audio-recorded, transcribed and translated into English. Transcripts were cross-checked by Ethiopian research co-ordinators for accuracy.

Findings from the participant observation were fed back to staff via a meeting to inform later stages of a wider project on health system strengthening. Minutes from this meeting, along with national guidelines relating to care for pre-eclampsia, were also used as data sources to triangulate findings and provide relevant context.

Clinical care guidelines in use at the time of the study included FMoH Management Protocol on Selected Obstetric Topics 2010, Standard Treatment Guidelines 2014 for Primary and General Hospitals and a Comprehensive Emergency Obstetric and Newborn Care Training Manual. The FMoH guidelines advocated for four focused ANC visits emphasising quality rather than quantity. Usual care for women with low-risk pregnancies was provided at health centres and the other visits conducted by health extension workers at health posts close to women's homes. Pregnant women identified to have risk factors including among others, previous pre-eclampsia, chronic hypertension, diabetes or renal disease were, according to national guidelines to be referred to higher level care or offered closer follow-up.

## Data analysis

We undertook thematic analysis using an inductive approach.[24] Field notes, interview transcripts and the minutes of the meeting were uploaded into NVivo12 software. Data were read and re-read, and initial codes were generated, reviewed and refined by TR, TEG and CH (see online supplemental file 1). All data were then recoded by TR and themes generated and reviewed with CH and RD. Themes and subthemes were checked against the data to examine relationships between them.

## Patient and public involvement

The design of this study focused on examining the lived experience of women affected by pre-eclampsia and their families. We included formal and informal interviews with women. Our research questions were informed by endeavouring to understand the priorities and experiences of patients. They were not involved in the planning, design or analysis of this study.

## RESULTS

Women and healthcare providers described significant systemic barriers to the provision of quality care. Basic structural necessities such a running water, soap and electricity were intermittently unavailable. This was particularly evident at the primary hospital. During 4 days of observation at the primary hospital, there was no running water. There were no sinks in any of the inpatient wards or clinical areas, including the infection prevention room at the primary hospital.

Our analysis revealed contextual challenges that women and professionals faced, within which they sought to receive or provide quality care for pre-eclampsia. We present three main themes: inconsistencies and inequity, improvisation and making do, and trust and support.

### Theme I: inconsistencies and inequity

Key guidelines identified by staff as informing care for pre-eclampsia at hospital level were acknowledged by some professionals as being outdated and were adhered to variably. Women admitted to the hospital with pre-eclampsia were often managed according to the national guideline but not consistently. Variability in clinical practice related to incomplete clinical assessments, including not eliciting or documenting significant risk factors such

as a history of eclampsia and previous stillbirth, and a lack of knowledge, for example, of established thresholds of severe hypertension.

> She will be given magnesium sulphate ($MgSO_4$) if her systolic BP is above 180 or diastolic above 110, we will give her hydralazine to drop the elevated BP. (Midwife, M11)

Guidelines and practice were ambiguous, relating to the management of antenatal women presenting with, or at risk of, eclampsia, who were preterm and whose BP normalised. Staff reported that women were discharged prior to delivery if their BP was controlled with antihypertensives.

> Q: She came with eclampsia; under what condition you will decide and discharge her?
>
> A: When her BP become normal, and her protein level reduces. It will become normal when we give them antihypertensive. There are many mothers who are discharged. (Midwife, M11)

Some medical staff reported they used diazepam to control seizures despite the availability of magnesium sulfate and guidelines stating this as the drug of choice.

Women described concerns about competence and safety. Some were particularly worried about the danger this posed to more vulnerable, less educated women who may be unable to advocate for themselves. A family member of a woman who died described conflicted feelings, both troubled by the care received while recognising providers' motivations were honourable.

> I have seen a lot there, we or you may read English a little and yes, your parents may read Amharic but how about those women who do not know anything. Yes, mistake a lot there, especially practitioners who provide medicine at resting room after surgery if they are not very careful, I do not know who will make them accountable. (Woman with previous pre-eclampsia, M28)

> In this hospital, they don't cooperate. We are grumbling because of them. We are blaming the hospital for her death…Yes, her body refused to circulate someone else's blood and get swelled….I am not better than the health professionals; they were trying to save her not to kill her…yes…I am thinking this was because I am hurt not because I am better than the health professionals. (Family of deceased woman, M26)

A limited understanding of the aetiology of pre-eclampsia was reflected in the advice given to women regarding their diagnosis.

> They used to tell me not to drink soft drinks, tea and the like. And that I don't have to use too much salt and need to decrease my fluid intake until the blood pressure goes down. (Woman with previous pre-eclampsia, M37)

Inconsistencies in triage of pregnant women related to a lack of skills, knowledge or experience of staff. Inappropriate triage was reported to result in a failure to recognise risk, escalate and manage complications appropriately.

> One of the things that needs improvement is developing the health professionals' skill and knowledge. Most of the problems are happening because of lack of knowledge. Most things are happening by not considering as a problem when actually problem has happened to mothers. Therefore, by improving the awareness we can change many things. (Senior clinician, M32)

> Sometimes triage might not be done appropriately, or due to limited knowledge and skill gap, the general practitioner might miss some cases. But once the surgeon is involved, he will be fully responsible. (Medical director, M49)

Relevant to the variation in clinical practice and competence were inadequate and inequitable access to training opportunities and professional development as described by providers. Some staff linked this to poor support from the hospital administration, while a senior clinician perceived a professional culture lacking in motivation to pursue independent study and continuing professional development.

> Most of the time, there are no trainings. The process of informing new information and participating the health professionals on training is very poor. This is one of the problem of administrators. Sometimes there are various types of training opportunities for their health professionals or the administrators will train the health professionals on areas where they believe there is a gap. They may also forbid the health professionals that was requested to get the training by paper from the high-level officials. This is the main problem of the administrators in our health institutions. The training does not have to be for getting the per diem rather it has to be used to updating the knowledge and to apply what you have got from the training. The way of administrators supporting this is very poor. (IESO, M48)

> The first thing is when there are such trainings, our society as a whole have a very poor reading habit. People don't sit down and read the guideline. (Medical director, M49)

Inconsistencies in how policies were implemented were also observed and reported. This variation may have been intended to ameliorate inequities in the system but also exacerbated them. National-level policy for free maternal care was implemented differentially dependant on clinical situation. In this study setting, women admitted to hospital with complications such as pre-eclampsia, requiring inpatient care but not delivery, were required to pay for services. Women attending purely for delivery received free care, whereas women with potentially

life-threatening complications requiring several medications, longer inpatient stays and at higher risk of delivering preterm infants in need of neonatal care, who were able to, were required to pay. However, women from low socioeconomic status households could be considered for exemption by the hospital management or could access free care if community-based health insurance had been rolled out in their area. Nonetheless, out-of-pocket payments for antihypertensive drugs were common. Financial barriers were observed to present tangible challenges for women with potentially serious complications related to pregnancy. Clinicians were not always aware of policy or how to apply it in practice.

### Case vignette

A recently postpartum woman presented to the emergency department with significant pitting oedema to bilateral lower limbs. She was reviewed by the doctor on-call, and laboratory investigations were requested. She attended the lab but was asked for 160 birr (£2.80), which she was unable to pay. She did not have a community-based health insurance card. The GP asked her to go back to her village to collect the insurance card, then to visit the local health centre and ask for referral back to the hospital for a full assessment. She did not return to the hospital that day.

> It is not free whether she is here or discharged. The hospital will fulfil things required for delivery, like gloves and others. The mothers may not buy the medication since they may not have money; there are mothers who don't have any money. Thus, even if we told them to buy it, they might not buy the medication… like antihypertensive, they have to pay…Yes, she has to purchase it. If methyldopa or so on is prescribed to her at ANC, she has to purchase it from her own pocket… It would have been good if it is possible to make laboratory services and medications prescribed for the mothers free as well. They would have come if it is free (Midwife, M11)

> The problem that we have in this hospital is, sometimes after deciding to admit them, there is some kind of problem on the payment system. For example, if a pregnant woman is close to due date and or going to give birth after some days and is going to be admitted, she is not going to pay. But if she has been admitted not for delivery, she would pay. There are these kinds of unclear rules. If it is free for one person, it should be free for all. (Senior clinician, M32)

### Theme II: improvisation and making do

Improvisation emerged as a means of managing or making do with limited resources, both material and human, within both hospitals. This centred around a lack of equipment, medications, staff and support. The barriers and facilitators to early detection, care escalation and appropriate management of pre-eclampsia are described in table 1.

BP monitors were lacking in both hospitals. Staff made do by sharing, negotiating use between departments or improvising with inadequate equipment such as a plastic training stethoscope used with a sphygmomanometer. At the primary hospital, there was no BP monitor available in the antenatal clinic. If midwives felt a BP check was needed, the woman was escorted to the ward to have this done.

Midwives at the general hospital made do with one semifunctioning cardiotocograph machine between six beds in the 'first-stage room' and two delivery beds. Midwives could only assess the current heart rate pattern or sit with women and observe in real-time, as paper was not available to record. Improvisation with regard to fetal monitoring for high-risk women was common and related to the lack of resources, but also a lack of knowledge of normal parameters and difficulties interpreting fetal heart rate patterns. Failure to recognise abnormal patterns was observed, which led to delays in taking action, either to expedite delivery or escalate. No fetal monitoring took place in the second stage of labour, regardless of earlier concerns. The reasons for this were difficult to ascertain, but this was an established culture of practice. On occasions, the management of suspected fetal distress at the general hospital involved the use of 40% dextrose infusions to the mother. These practices may have been related to a lack of specific guidance for midwives on the management of fetal distress.

Assessment of antenatal women presenting with reduced fetal movements close to term was inadequate. The fetal heart was auscultated to confirm presence. Women were not offered BP or urine screening, assessment of symphysial fundal height or a growth scan. At the primary hospital, no continuous fetal monitoring was available, and a diagnosis of presumed fetal distress was made following auscultation of decelerations using a fetoscope.

Precarious practices were observed, such as the use of fundal pressure during the second stage of labour when providers lacked confidence, the necessary skills or support to perform assisted delivery.

Families offered frequent support to plug gaps in the system, collecting patient notes, requesting blood tests, clearing buckets of urine and blood on the labour ward, procuring drugs and equipment from private pharmacies when not available in labour, and providing support for women in labour when allowed.

### Theme III: trust and support

Communication skills and styles varied between individuals with examples of kind, compassionate interactions, professionals checking women's understanding and providing reassurance. However, a culture of informed consent and shared decision making was not widespread. Women were frequently subjected to vaginal examinations in an open ward without privacy. There were observed examples of abrasive interactions and failures to take women's concerns seriously. Decisions

**Table 1** Barriers and facilitators to early detection, care escalation and appropriate management of pre-eclampsia in hospital

| Hospital-level care for pre-eclampsia | Barriers | Facilitators |
|---|---|---|
| Early detection and recognition of risk | ▶ A lack of continuity of care and linkage between general outpatient department clinics, where women with chronic hypertension, diabetes or renal disease may be managed, and high-risk ANC clinic.<br>▶ Unclear guidelines for midwives, IESOs and doctors regarding outpatient management of women referred from lower levels of care with risk factors for pre-eclampsia.<br>▶ A lack of BP monitors in clinical areas where pregnant women were cared for. | ▶ Regular BP monitoring was observed for women admitted with severe hypertension (although not always documented). |
| Escalation of care | ▶ Staff responsible for triaging pregnant women were not always competent, experienced or supported enough to escalate care appropriately.<br>▶ Poor documentation of clinical history, risk factors, examination and care, given challenged communication between staff and different levels of the care pathway. | ▶ Referrals from lower levels of care (health centres and primary hospital) were reported to be frequent (although reported by staff as not always necessary).<br>▶ A clear professional hierarchy was described by staff, indicating they knew who they could refer on to. |
| Appropriate management | ▶ National guidelines were not evidence-based or up to date or were unclear, recommending precarious or potentially unsafe practices.<br>Management of hypertension (explicit advice not to treat non-severe hypertension ≥140/90–159/109 mm Hg, lack of timely management of severe hypertension, advice to wait 1 hour).<br>Timing of delivery (recommend delivery when cervix favourable for 'mild pre-eclampsia', advise avoiding post-term pregnancy but no explicit recommendation for delivery at 37 weeks).<br>▶ A lack of fluid balance monitoring was observed, and injudicious use of intravenous fluids concurrently with furosemide without a clear indication.<br>▶ Timing of antenatal steroids for fetal lung maturation (given to women with preterm pregnancy and hypertension even when delivery was not planned).<br>▶ Unclear and inconsistent processes for induction of labour, increasing the risk of caesarean section.<br>▶ Poor documentation of clinical care led to a lack of transparency and accountability.<br>▶ Minimal counselling regarding future risks including in subsequent pregnancies, increased risk of cardiovascular disease and stroke, and no guideline to support this practice. | ▶ Magnesium sulfate was available, and doses prescribed according to national and international guidelines.<br>▶ Guidelines recommend delivery within 24 hours of onset of symptoms in 'severe pre-eclampsia' and within 12 hours of eclampsia. |

ANC, antenatal care; BP, blood pressure; IESO, integrated emergency surgical officer.

about termination of pregnancy for women with early onset pre-eclampsia were made without explanation to women and were not documented in hospital notes. These interactions were described as 'oral reviews'. Staff were frequently observed carrying out clinical activities without documenting including performing vaginal examinations, ward rounds and administering drugs. Some staff reported this related to care being given outside prescribed protocols, for example, the timing of examinations during labour. Poor-quality documentation was also linked to a punitive culture and fear of being blamed for adverse outcomes. Strategies to improve documentation were in progress, including a documentation quality audit.

Allowing women family support during labour was inconsistent and limited by structural constraints such as the lack of privacy and space in the labour ward. However, relating back to the initial theme, inequitable application of the rules left women who were less able to advocate for themselves alone and lacking support, while others were allowed family members with them. Staff talked about the challenges they felt affected their ability to provide respectful care including workload and not feeling respected themselves.

**Table 2** Theme III: trust and support

| Theme III: trust and support | Verbatim quotes |
|---|---|
| Women experienced or witnessed lack of respectful care or staff attitudes that presented barriers to quality care | But there are also doctors who think they know everything, but they are below doctors. To your surprise, you see it many times, those who insult, who say you did not follow here so go away. But it has no problem. She enters for delivery after signing for the risk, right. She will be responsible for everything that could happen to her. But saying go away while she is about to give birth and because she did not follow there, I have seen a lot. (Woman with previous pre-eclampsia, M28) |
| Communication with women affected, what they understood about their diagnosis, and their trust in professionals and the care received | I am still not sure whether I do have or don't have the hypertension at that time. Finally, at the hospital I wasn't so sure whether I have hypertension or not but at the hospital, they said, "It is very much elevated, and she might get into a bad condition if we are waiting more than this". Then they made me sign and performed the surgery and I gave birth. That's all about it. I didn't understand whether they did that purposely to intimidate me or whether I exactly do have the hypertension. (Woman with previous pre-eclampsia, M34) |
| Doubt and anxiety about their diagnosis led to some women searching for alternative opinions. | Yes, it was at the hospital. I went three places to check my blood as a matter of fact… they told me to wait and I went to other places in the meantime to get my BP checked… When I checked there, the number was the same. So, I returned back to the hospital because I was told to remain there. I was very nervous. (Woman with previous pre-eclampsia, M25) |
| Staff felt challenged to provide compassionate and respectful care. | Due to the labour pain labouring mothers are aggressive sometimes which makes our job a little bit complicated while giving the service. At this moment it is difficult to practice the CRC [compassionate respectful care] as we hoped to. But you have to minimize your own issue and try to help the mother and her child. We should take care of the child regardless of the situation and the protocol sometimes. (Head midwife, M33) |
| Women reported staff were dissatisfied, and this affected quality of care and communication. | I hated many of the women. I am busy and I work here unwillingly [the midwife said]. It is her work, right. She gets paid, right. No one is doing one's job happily. She has to communicate peacefully even if she doesn't like the woman's blood or waste. (Woman with previous pre-eclampsia, M28) |
| Staff did not feel supported but were motivated to work by a sense of duty. | Support is very low. You work for your conscience. There are a lot of problems on the administration side. You just work because of feeling responsible. (IESO, M47) |
| Staff reported the need for more feedback as well as support to improve retention. | Giving recognition for the tasks that they have accomplished well. Appreciating or congratulating on a job well done and give constructive criticism when the work lacks will be great. If not, that person will be compelled to seek other more satisfying work environments. (IESO, M47) |

There were examples of low staff satisfaction reported by women affecting quality of care. Staff reported low level of professional support from the hospital administration and a sense of moral duty driving their commitment to work. Suggestions about how to support and retain staff reflected a need to develop more nurturing systems with opportunities for feedback to clinicians, both positive and negative. The theme of trust and support is further illustrated in table 2.

## DISCUSSION

This study found the provision of quality, person-centred care for pre-eclampsia in two Ethiopian hospitals was challenged by numerous systemic barriers. A lack of consistency in availability of resources, support and how policies were applied exacerbated inequities in care. This was particularly evident in our examples where women required admission and investigations for complications during pregnancy or the postnatal period but were unable to pay. Understanding structural drivers of poor-quality maternity care and mistreatment is vital to ameliorating the detrimental impact on future help-seeking.[25 26] There is a need to strengthen education and support for staff to provide respectful, person-centred care as well as addressing resource shortages.[27–29]

Variations in clinical competence, knowledge and experience meant staff providing maternity care were not always able to recognise risks or act on them. Outdated, ambiguous guidelines and inequitable opportunities for professional development exacerbated clinical uncertainty and practice not based on evidence. National guidelines specify against the use of antihypertensives for non-severe hypertension, and many women with significant hypertension were not treated. Inadequate documentation did not always reflect clinical risks, drugs administered or care given. Poor-quality documentation was linked to the provision of care divergent from protocol and fear of blame for poor outcomes. A lack of transparency, accountability and systems for support was observed and reported. Improvisation was used to address shortages in resources and a lack of clear guidance. This affected quality of care, particularly with regard to fetal

monitoring and neonatal care. Staff reported a need for more support, particularly from hospital management, and training opportunities.

## Strengths and limitations

This study offers an ethnographic account of care for women with pre-eclampsia in hospital in rural Ethiopia. Observational data, triangulated with in-depth interviews alongside relevant health policy and clinical guidelines, enabled a holistic examination of processes and quality of care and the factors challenging them. We explored diverse perspectives including women affected by pre-eclampsia, families of women who did not survive, healthcare providers and managers. This approach allowed a deeper understanding of the context of care including clinical, sociocultural and bureaucratic aspects. Our analysis was informed by participants' descriptions and recollections of care they gave or received, as well as observed explicit and tacit actions and interactions. We used the guidelines to contextualise our observations and the accounts of care.

Conducting a rapid ethnography offered both advantages and disadvantages. We were able to feedback key practical findings from our observational work to staff and district-level stakeholders, in a timely way, informing quality improvement (QI) initiatives as part of wider health system strengthening efforts. This also offered an opportunity for member checking, ensuring our observations reflected real-world challenges faced by providers. Although longer periods of observation may have revealed further useful insights, recurring themes were evident in field notes. Another limitation was the need for translation for one observer. English was used by staff to document in hospital notes, and all staff spoke English; however, interactions with patients and between staff were usually in Amharic. Subtle meaning or nuance may have been lost in translation or interactions misinterpreted. Furthermore, the 'outsider' status of a non-Ethiopian team member is likely to have affected behaviour of participants. However, this is a recognised challenge when conducting ethnography and was considered when interpreting findings.[30 31] Staff were friendly, cooperative and keen to share their experiences, and often asked observers for advice or feedback.

## Interpretation

In our study early detection, care escalation and appropriate management for pre-eclampsia were undermined by a lack of up-to-date guidance based on evidence. Clinical manifestations and conceptualisations of pre-eclampsia are variable. The International Classification of Diseases has adopted an updated, broader definition of pre-eclampsia to include hypertension and proteinuria or new-onset maternal organ dysfunction, neurological sequelae or fetal growth restriction.[9 32] Inconsistency remains in national and international clinical practice guidelines implementation of the broad definition and the categorisation of 'severe' pre-eclampsia or pre-eclampsia

with severe features.[33] This is important as guidelines for management and recommendations for interventions are linked to classification of the disorder. These inconsistencies were mirrored in the care we observed in this study.

Early prediction and prophylaxis for pre-eclampsia were not a part of routine care in our study site, nor recommended in Ethiopian guidelines. Most Ethiopian women consume insufficient calcium, but large regional variation exists.[34] Implementation of interventions requiring early contact may be challenging in a context where first trimester care is rare, particularly at levels of the system where providers are competent and confident to prescribe. However, policy makers should consider evidence-based guidance and international recommendations to support prophylaxis and reduce burden of disease in at-risk women.[11 35 36] The updated 2020 national guideline does not address these discrepancies.

Accurate measurement of BP is fundamental to detecting and managing pre-eclampsia. In our study sites, both fully automated devices and mercury sphygmomanometry were used. The latter requires skills and training to ensure accuracy. The haemodynamic adaptations in pregnancy and pre-eclampsia affect the oscillometric measurement used in automated devices. Few automated devices have been validated in pregnancy or pre-eclampsia. The use of unvalidated or uncalibrated automated devices may lead to significant inaccuracies in BP measurement.[37] Accurate and cost-effective devices have been validated in pre-eclampsia and are useful in health systems where low-skilled staff are involved in maternal care.[38–40]

Triage, recognition of risk and care escalation for pre-eclampsia were identified as areas for improvement in our study. In settings where resources are limited, risk prediction models may offer support to staff to target interventions and decision making around place of care.[41] Our findings concur with previous studies reporting gaps in knowledge and practices required for early detection and management of pre-eclampsia in facilities in low-resource settings.[42] Health system constraints, training, supervision and motivation of staff affect clinical performance and outcomes.

Health planners must focus on ensuring that funding in maternity services and the allocation of resources work to address current challenges to equity. This requires understanding contexts of care and how policies are implemented at a local level. Improvements in external factors such as guidelines and infrastructure must be mirrored by changes in cultures of care that support a shared understanding of quality and focus on team competencies and supportive leadership. Previous QI initiatives in Ethiopia have failed due to lack of understanding and adaptation to local contexts.[43] Understanding formal and informal power structures, considering spheres of influence and creating social and reputational incentives for change may improve focus and engagement. A study aiming to measure and improve quality of intrapartum care showed that having lead hospitals demonstrate the feasibility of

their assessment tool helped to promote its wider acceptance, as other hospitals sought to achieve the national recognition given to the lead hospitals.[44] Participatory approaches that bring together women, clinical and managerial staff may help to shift power dynamics where unconstructive hierarchies exist. Developing shared goals, strategies to achieve them and clarifying roles early on can improve sustainable and effective implementation.

## CONCLUSIONS

Quality care for pre-eclampsia requires organisational change to create safe spaces for learning and improvement, alongside efforts to put women at the centre of care and ensure providers are equipped with knowledge, resources and support to adhere to evidence-based practice.

National guidelines for HDP including pre-eclampsia should recommend treatment of BP consistently ≥140/90 mm Hg to reduce the risk of developing severe hypertension and other complications and to urgently treat severe hypertension ≥160/110 mm Hg.[9] They should also include guidance for professionals on counselling women about their future risks, including of having a subsequent pregnancy affected by pre-eclampsia, cardiovascular disease and stroke, and strategies to reduce these. As the growing burden of non-communicable disease emerges globally, opportunities should be seized while women are engaged with health services to optimise longer-term outcomes.

More research is required to understand whether evidence established in high-resource settings is applicable in LMICs. This should include implementation research and cost-effectiveness of early prediction and prevention strategies.[36]

It is acknowledged that staff are working under difficult conditions within resource-constrained systems. This study allows a focus for where to instigate system change to improve the management of pre-eclampsia for women and support staff to deliver quality care. This should include improvements in antenatal screening and detection of hypertension.

### Author affiliations
[1]Department of Women and Children's Health, King's College London, London, UK
[2]Department of Women and Children's Health, School of Life Course Sciences, King's College London, London, UK
[3]Centre for Innovative Drug Development and Therapeutic Trials for Africa (CDT-Africa), Addis Ababa University College of Health Sciences, Addis Ababa, Ethiopia
[4]Department of Obstetrics and Gynaecology, Addis Ababa University College of Health Sciences, Addis Ababa, Ethiopia
[5]Department of Global Health and Social Medicine, King's College London, London, UK
[6]Institute of Psychiatry, Psychology and Neuroscience, Health Service and Population Research Department, Centre for Global Mental Health, King's College London, London, UK

**Acknowledgements** We gratefully thank all participants for their time and contributions to the study. We thank Sewit Hessebon, Hanna Negussie and Medhanit Getachew.

**Contributors** TR: conceptualisation, methodology, investigation, data curation, formal analysis, visualisation, writing (original draft, reviewing and editing), guarantor. AS: conceptualisation, funding acquisition, supervision, writing (reviewing and editing). JS: conceptualisation, funding acquisition, methodology, supervision, writing (reviewing and editing). TEG and RD: investigation, formal analysis, writing (reviewing and editing). AA: conceptualisation, funding acquisition and writing (reviewing and editing). RM and CH: conceptualisation, methodology, funding acquisition, supervision, writing (reviewing and editing).

**Funding** This research was funded by the National Institute of Health Research (NIHR) Global Health Research Unit on Health System Strengthening in sub-Saharan Africa, King's College London (GHRU 16/136/54), and CH was supported by an NIHR RIGHT grant (NIHR200842) using UK aid from the UK government. The views expressed are those of the authors and not necessarily those of the NHS, the NIHR or the Department of Health and Social Care. CH receives support from AMARI as part of the DELTAS Africa Initiative (DEL-15-01). JS is an NIHR senior investigator and is supported by the NIHR Applied Research Collaboration South London at King's College Hospital NHS Foundation Trust.

**Competing interests** None declared.

**Patient and public involvement** Patients and/or the public were not involved in the design, conduct, reporting or dissemination plans of this research.

**Patient consent for publication** Not applicable.

**Ethics approval** This study involves human participants. All methods were carried out in accordance with relevant guidelines and regulations, including the Declaration of Helsinki. Ethical approvals were granted by Addis Ababa University (026/18/Psy) and King's College London (LRS-17/18-5526). Institutional-level consent was given by hospital management for observational data collection. Interview participants were required to give individual consent. The participants gave informed consent to participate in the study before taking part.

**Provenance and peer review** Not commissioned; externally peer reviewed.

**Data availability statement** Data are available upon reasonable request. Availability of data and materials The data generated and analysed during this study are not publicly available to protect individual anonymity but may be available from the corresponding author on reasonable request.

### ORCID iDs
Tanya Robbins http://orcid.org/0000-0001-7851-7171
Charlotte Hanlon http://orcid.org/0000-0002-7937-3226

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
