## [Reviewer comments · BMJ Open]

ARTICLE DETAILS

TITLE (PROVISIONAL)	Understanding challenges as they impact on hospital level care for pre-eclampsia in rural Ethiopia: a qualitative study
AUTHORS	Robbins, Tanya; Shennan, Andrew; Sandall, Jane; Eshetu Guangul, Tigist; Demissew, Rahel; Abdella, Ahmed; Mayston, Rosie; Hanlon, Charlotte

VERSION 1 – REVIEW

REVIEWER	Atkins, Bethany University College London
REVIEW RETURNED	06-Apr-2022

GENERAL COMMENTS	Well done on a very good paper. This is an interesting study which describes some of the challenges faced in providing good quality care for women with pre-eclampsia in Ethiopia. Please see the attached document for comments, as well as some comments below. Abstract - excellent Introduction - very good. I think a greater acknowledgement of the uncertainty of some of the estimates of pre-eclampsia and eclampsia in LMICs should be included, with perhaps a mention of the increasing burden of NCDs (ie essential hypertension) Methods - I would like to see some more detail regarding the interview methodology. Were these structured, semi-structured, unstructured? How were staff members, women and family members identified and recruited? Who was conducting the interviews and what was their relationship to the interviewees (eg as a researcher were they known to the clinicians etc). I cannot see a reporting checklist eg SRQR or COREQ Results - the results section is comprehensive and clearly presented. I like how quotes have been used to illustrate the findings. Discussion - I feel the discussion would benefit from adding a section reflecting on the impact of limited resources and funding in this context, as this is described in the results, and is an important finding. For example, inconsistencies in women being charged for hospital admission, women advised on investigations but unable to pay for them (case vignette). Additionally, I think a comment on the detrimental impact of poor quality care on future care attendance is important.
--

REVIEWER	Dempsey, Amy Population Council
REVIEW RETURNED	12-Apr-2022

GENERAL COMMENTS	It's an interesting paper, but revising the structure would be beneficial. As written, it's quite cluttered and confusing, primarily the results section. Reorganizing the different view points would fix that. Reviewer also provided an attachment with comments – please contact the publisher to view this.
--

VERSION 1 – AUTHOR RESPONSE

Reviewer 1

Well done on a very good paper. This is an interesting study which describes some of the challenges faced in providing good quality care for women with pre-eclampsia in Ethiopia.
Please see the attached document for comments, as well as some comments below.

Abstract - excellent

Comment 1:

Introduction - very good. I think a greater acknowledgement of the uncertainty of some of the estimates of pre-eclampsia and eclampsia in LMICs should be included, with perhaps a mention of the increasing burden of NCDs (i.e. essential hypertension)

Response:

We have added this sentence

“ In addition to these uncertain estimates, there is an increasing burden of non-communicable diseases in LMICs. These include chronic hypertension and diabetes which are known to increase the risk of developing pre-eclampsia.” (pg 5, line 89 - 91 - tracked changes)

Post-it note comment:

I believe this is true, but this reference doesn't evidence this.

Response:

Error in referencing corrected. (pg 5, line 86)

Comment 2:

Methods - I would like to see some more detail regarding the interview methodology. Were these structured, semi-structured, unstructured? How were staff members, women and family members identified and recruited? Who was conducting the interviews and what was their relationship to the interviewees (e.g. as a researcher were they known to the clinicians etc).
I cannot see a reporting checklist e.g. SRQR or COREQ.

Response:

We have already described this (see as follows):

21 formal in-depth semi-structured interviews were conducted. We purposively sampled 10 staff members (two midwives, three IESOs, four doctors including an obstetrician and a district health manager), eight women with lived experience of pre-eclampsia and three family members of women who had died due to pre-eclampsia. Interviews were conducted in Amharic, audio-recorded, transcribed and translated into English. (pg 7, line 161 -167)

We have added the following sentence:

“ Women affected by pre-eclampsia were identified through routinely collected data. The research team conducting interviews were not known to interviewees.” (pg 8, line 164 - 165)

COREQ checklist included and page numbers updated as per Editor’s comment.

Results - the results section is comprehensive and clearly presented. I like how quotes have been used to illustrate the findings.

Post-it note comment:

Is this comment from senior or junior clinical staff? Is it self-reflective, or is it senior staff stating that other staff are lacking motivation as suggested in the discussion?

Response:

We have amended as below:

“Some staff linked this to poor support from the hospital administration, whilst a senior clinician perceived a professional culture lacking in motivation to pursue independent study and continuing professional development.” (pg 12, line 128)

Post-it note comment:

Is it just confidence? Or training, skills, support etc?

Response:

We have amended as below:

“Precarious practices were observed such as the use of fundal pressure during the second stage of labour when providers lacked confidence, the necessary skills or support to perform assisted delivery.” (pg 16, line 379)

Post- it note comment:

I think you are contrasting potentially unreliable fully automated devices with validated semi-automated devices, but I did not find this paragraph clear.

Response:

No, we are not contrasting fully automated with semi-automated devices. We state unvalidated automated devices may lead to inaccuracies.

To ensure clarity, we have amended as follows:

“Accurate measurement of BP is fundamental to detecting and managing pre-eclampsia. In our study sites both fully automated devices and mercury sphygmomanometry, were used. The latter requires skills and training to ensure accuracy. The haemodynamic adaptations in pregnancy and pre-eclampsia affect the oscillometric measurement used in automated devices. Few automated devices

have been validated in pregnancy or pre-eclampsia. The use of unvalidated or uncalibrated automated devices may lead to significant inaccuracies in blood pressure measurement.” (pg 22, line 490 - 495)

Comment 3:

Discussion - I feel the discussion would benefit from adding a section reflecting on the impact of limited resources and funding in this context, as this is described in the results, and is an important finding. For example, inconsistencies in women being charged for hospital admission, women advised on investigations but unable to pay for them (case vignette).

Additionally, I think a comment on the detrimental impact of poor-quality care on future care attendance is important.

Response:

In response to the reviewer, we have stressed the impact of these findings by adding the following sentence:

“This was particularly evident in our examples where women required admission and investigations for complications during pregnancy or the postnatal period but were unable to pay. Understanding structural drivers of poor-quality maternity care and mistreatment is vital to ameliorating the detrimental impact on future help-seeking. There is a need to strengthen education and support for staff to provide respectful, person-centred care as well as addressing resource shortages.” (pg 19, line 425 - 430)

Post-it note comment:

I think that the discussion section is missing a section focusing on resourcing and funding of the health system, and its impact as described in the results section.

Response:

We wanted to emphasise the point that although clearly more funding and resources are needed, these must be matched by shifts in internal systemic factors.

We have added the following:

“Health planners must focus on ensuring that funding in maternity services and the allocation of resources work to address current challenges to equity. This requires understanding contexts of care and how policies are implemented at a local level.” (pg 23, line 512 - 514)

Post-it note comment:

I agree with this, but I don't think this is supported by your study results.

Response:

We did find evidence to support this statement in our study as described in our methods. Our study included a document review of national guidelines informing care. These are unclear about the threshold at which to initiate treatment. Current evidence supports tight blood pressure control <135/85mmHg to reduce the risk of severe hypertension and associated adverse outcomes. We hope you agree and have not changed the conclusion.

Reviewer 2

Comments to the Author: (see attached file)

It's an interesting paper but revising the structure would be beneficial. As written, it's quite cluttered and confusing, primarily the results section. Reorganizing the different viewpoints would fix that.

Response:

The first reviewer has commented "the results section is comprehensive and clearly presented. I like how quotes have been used to illustrate the findings." We hope you agree we shouldn't make a substantive change to structure but we have looked at this critically given reviewer 2's comment. In response we have removed table 1 and incorporated the quotes into the main body of text and hope this improves the structural concerns of this reviewer.

Comment 1:

Why here? Were other zones considered? Who participated in the decision making for choosing this location?

Response:

We have added the following sentence:

"This study setting was selected to include a predominantly rural population, spread across lowland and highland areas. This is to provide generalisability of our findings. Geographical barriers in the area are significant and access to ambulances limited." (pg 6, line 125 - 127)

Comment 2:

Is there a difference between maternal mortality in rural vs. urban settings? May want to address - here or above - how health challenges in rural settings impact maternal health outcomes.

Response:

There is no robust data disaggregating maternal mortality in Ethiopia by rural or urban settings. Accessing care is likely to be more challenging in a rural area, where most deliveries occur. This is acknowledged above.

Comment 3:

You may want to go into more detail about these formal and informal interviews. How were the formal interviews organized? How do you define "informal"?

Response:

We have added the following sentence in the methods section (rather than in PPI section).

"Informal interviews were conducted with staff to enquire about specific aspects of care or interactions during participant observation where further clarification or context was sought." (pg 7, line 158 -159)

Comment 4:

The tables are somewhat confusing. These quotes would be more engaging if they were incorporated into your prose. If the tables stay, it would be better to have separate tables for the types of speakers, as opposed to one large table.

Response:

See above response. We have removed table 1 and incorporated the quotes into the main body of text and hope this improves the structural concerns of this reviewer.

Comment 5:

You could address this more in the study setting or intro sections and explain some of the social influences that impact early and regular ANC patterns.

Response:

This paper is focused on secondary level care for pre-eclampsia rather than routine ANC and screening for pre-eclampsia.

Comment 6:

This section is more about recommendations than it is a conclusion.

Response:

The BMJ Open Guidelines to Authors recommends that this section includes primary conclusions and their implications, suggest areas for further research if appropriate.

Comment 7:

Recognizing this paper is focusing on secondary hospitals, it may benefit the paper to address the importance of ANC and how it should be used to detect hypertension before complications arise.

Response:

We have added the following sentence to the conclusions:

“This should include improvements in antenatal screening and detection of hypertension.” (pg 24, line 546 - 547)

VERSION 2 – REVIEW

REVIEWER	Atkins, Bethany University College London
REVIEW RETURNED	02-Sep-2022
GENERAL COMMENTS	Please see the attached document (please contact the publisher to view this) for a small number of comments. Overall I think the paper would be acceptable with only very minor changes.
REVIEWER	Dempsey, Amy Population Council
REVIEW RETURNED	10-Jan-2023
GENERAL COMMENTS	No further comments.

VERSION 2 – AUTHOR RESPONSE

Reviewer 1: The statement about NCD needs referencing.

Response: This has now been referenced.

Reviewer 1 : I am not sure what this first quote is communicating and whether it adds anything that is not already written in the main text.

Response : This quote has been left in the manuscript as we feel it's important to include a range of participants voices, particularly non-medical cadres providing emergency obstetric care who may have less access to training opportunities.

Comments to the Author:

Please see the attached document for a small number of comments. Overall I think the paper would be acceptable with only very minor changes.

Reviewer: 2

Amy Dempsey, Population Council

Comments to the Author:

None!